# Theoretical modelling of wakes from retractable flapping wings in forward flight

Ben Parslew and William J. Crowther

School of Mechanical, Aerospace and Civil Engineering, The University of Manchester, UK

## ABSTRACT

A free-wake method is used to simulate the wake from retractable, jointed wings. The method serves to complement existing experimental studies that visualise flying animal wakes. Simulated wakes are shown to be numerically convergent for a case study of the Rock Pigeon in minimum power cruising flight. The free-wake model is robust in simulating wakes for a range of wing geometries and dynamics without requiring changes to the numerical method. The method is found to be useful for providing low order predictions of wake geometries. However, it is not well suited to reconstructing 3d flowfields as solutions are sensitive to the numerical mesh node locations.

## INTRODUCTION

Fluidic forces on flying animals have been derived from direct measurements of pressure on the wing and tail (*Usherwood, Hedrick & Biewener, 2003*; *Usherwood et al., 2005*). Flow visualisation is an alternative technique that allows forces to be inferred through examining the structure of the wake (e.g., *Lauder & Drucker, 2002*; *Spedding & Hedenström, 2008*; *Taylor, Triantafyllou & Tropea, 2010*) this approach aims to reduce interference between the subject and measurement apparatus, allowing the experimenter to capture more natural kinematics.

To complement experimental measurements, theoretical models of wakes have been developed that predict flying performance. Early examples include models of flapping-wing flight that prescribe the wake geometry as a series of vortex rings (*Rayner, 1979a*; *Rayner, 1979b*; *Rayner, 1987*; *Ellington, 1984*). However, prescribed-wake models are limited by their inability to capture evolution of the wake geometry over time. More recently, Navier-Stokes computations have been used to simulate the flow around flapping wings (*Liu et al., 1998*). While this approach calculates properties of the entire flowfield, there are drawbacks in terms of computational cost. Also, considerable effort is required to construct the numerical mesh that discretizes the flowfield.

Free-wake methods developed for aerospace applications simulate wakes from lifting surfaces without resolving the entire flowfield (*Smith, Wilkin & Williams, 1996*; *Parslew, 2012*; *Bagai & Leishman, 1995*; *Tarascio et al., 2005*; *Stock, 2006*). These methods calculate

Corresponding author
Ben Parslew,
ben.parslew@manchester.ac.uk

**Peer**J _______________

induced velocity from vorticity in the wake, which arises due to the presence of a lifting surface. As a result, they implicitly predict the evolution of the wake geometry as it propagates downstream. Free-wake models are not restricted to specific lifting surface kinematics, and are therefore applicable to fixed, rotary and flapping wings.

The contribution of this paper is to apply the free-wake methodology to retractable, jointed, flapping wings. A parsimonious approach will be used, whereby a simple but extensible model will be developed using a minimal number of input parameters. The model will capture the fundamental wake geometry, and can later be extended to simulate more complex fluidic phenomena such as vortex viscous core growth and vortex stretching. The primary aim of the study is to demonstrate numerical convergence of the model for a case study of the Rock Pigeon, *Columbia livia*, in cruising flight; numerical convergence will be recognised as the tendency of the simulation to approach a fixed solution as the resolution of the numerical scheme increases. Preliminary findings for simulated wake geometries will be presented for different wing geometries and dynamics.

## METHOD

### Free-wake model

The free-wake model presented here is derived from a model used for analysing helicopter rotor wakes (*Bagai & Leishman, 1995*; *Parslew & Crowther, 2010*). Vortex filaments are assumed to be shed from the trailing edge of the wing (Fig. 1). Filaments are discretized into a series of segments between Lagrangian markers. Markers move with the local flow velocity, transporting flow properties with them. Modelling the flow as inviscid and irrotational, Helmholtz's second law of vortex motion implies that vortex filaments move as material lines, and thus the velocity of any marker is equal to the local flow velocity:

$$\frac{d\mathbf{x}}{dt} = \mathbf{V};$$
(1)

$\mathbf{x}$ is the marker position and $\mathbf{V}$ is the local fluid velocity, which is the vector sum of the freestream velocity, $\mathbf{V}_\infty$ (due to motion of the body through the fluid), and the velocity induced from vortex filaments, $\mathbf{V}_I$. The main challenge in solving Eq. (1) is calculating the vortex induced velocity. This is achieved using the Biot-Savart law, which forms the foundation of the model (see section induced velocity).

The free-wake model can be implemented using three different approaches (Figs. 1A–1C). Horseshoe vortex models assume a bound vortex on the wing and a pair of trailing vortex filaments shed from the trailing edge. The simplest model assumes a single horseshoe vortex with trailing filaments shed from the wingtips (Fig. 1A). Multiple horseshoe vortices can be used to model the effects of non-uniform lift distributions on the wing that occur during flapping-wing flight (Fig. 1C).

The vortex ring model (Fig. 1C) is similar to the trailing vortex model, but also takes into account spanwise vorticity shed due to temporal changes in lift. Shed vortex segments are calculated as line segments between markers shed at the same time on adjacent filaments. Circulation of these segments is given as the change in bound circulation on

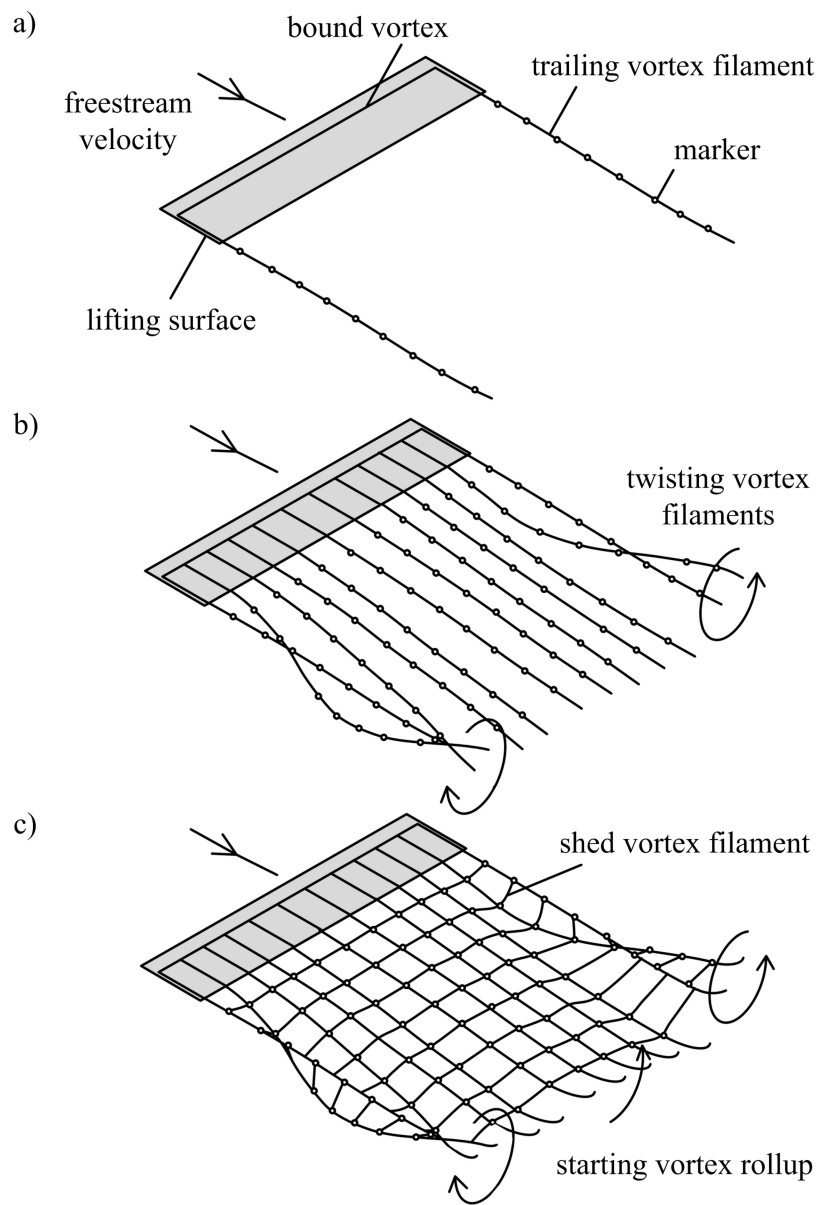

**Figure 1 Free-wake models.** (A) Single horseshoe vortex model; (B) multiple horseshoe vortices model; (C) vortex ring model. Horseshoe vortex models include the bound vorticity on the lifting surface and trailing vorticity in the wake. In all models the wake downstream of the wingtips rolls up, and this is visible in models (B) and (C) as the trailing vortex filaments twist. The vortex ring model (C) also includes the transverse vorticity component that is shed from the surface due to temporal changes in lift; when the surface is accelerated from rest the transverse vorticity component leads to a starting vortex.

the wing over the time interval between the release of successive markers (*Tarascio et al., 2005*). The circulation on a segment between a pair of markers is the vector sum of the circulation from the two adjacent vortex rings on which the markers lie (Fig. 2B). The vortex ring model is selected for the present study to capture spanwise vorticity that arises in flapping-wing wakes due to the temporal variation in lift throughout the wingbeat.

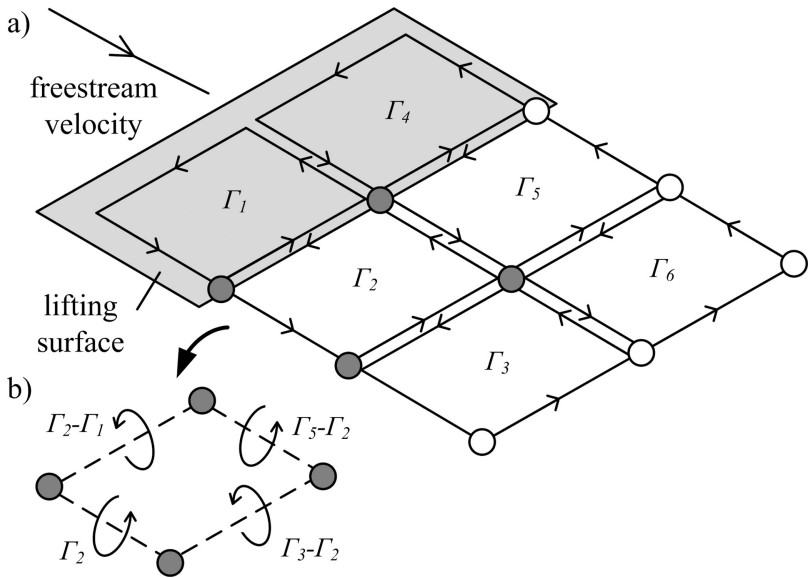

**Figure 2  Illustration of vortex rings being shed from a lifting surface.** (A) Lagrangian markers define the ring locations. Circulation ($\Gamma_{1-6}$) is summed for neighbouring vortex rings to give the net circulation on a segment between each marker pair. (B) Example of net circulation on four filament segments.

## Induced velocity

Velocity induced on a marker located at $\mathbf{x}_m$ due to the presence of the $n$th filament segment is given from the Biot-Savart law:

$$V_I = \frac{\Gamma^{(n)}\mathbf{d}^{(n)} \times (\mathbf{x}_m - \mathbf{x}^{(n)})}{4\pi \,\|(\mathbf{x}_m - \mathbf{x}^{(n)})^3\|};$$ (2)

where $\Gamma^{(n)}$ and $\mathbf{d}^{(n)}$ are the circulation and length of the $n$th filament segment and $\mathbf{x}^{(n)}$ is the location of the $n$th segment marker. Typically, a vortex core model is used to avoid numerical singularities when the distance between the marker and filament segment tends to zero (*Stock, 2006*). However, in the present work all simulations were found to be stable and convergent without using a viscous vortex core model.

The contribution of induced velocity from each filament segment is calculated using Eq. (2) and summed to give the net velocity on a marker. The procedure is repeated for all markers, forming an $n$-body problem. This method of direct evaluation is only feasible for small numbers of markers and could be accelerated by long-range cutoff approximations (*Stock, 2006*); as the present algorithm has not been optimised for performance the computational cost increases quadratically with the number of markers, and is not representative of the cost of most point vortex simulations.

The local lift per unit span on a surface, $L'$, is calculated using blade-element theory (*Parslew, 2012*; *Parslew & Crowther, 2010*); while this approach only captures quasi-steady
aerodynamic lift the wake model can also be used with methods that include unsteady effects. The Kutta-Joukowski theorem gives the circulation around the wing as

$$\Gamma = \frac{L'}{\rho \|\mathbf{V}_\infty\|}. \tag{3}$$

The instantaneous circulation is calculated for each blade-element, giving the shed and trailing circulation of filament segments shed from the trailing edge at a given time. Equation (2) is then evaluated for all markers in a time-marching simulation. Marker locations are updated using a one-step Euler integration scheme for the first timestep, and a two-step Adams-Bashforth scheme for all subsequent timesteps.

### Limitations

- Induced velocity predicted by the free-wake simulation is not included in the blade-element model. As the freestream velocity magnitude is typically ten times that of the induced velocity, this approximation is not believed to significantly affect the simulated wake geometry.

- The Kutta condition is not enforced in the model, meaning that the velocity field immediately downstream of the lifting surface trailing edge will not be accurately resolved. This factor will be most significant when the angle attack is high, such as in slow flight and hover. Despite this, the model is still useful for making low-order predictions of the wake geometry as it propagates downstream.

- As an inviscid flow model, the wake dynamics are driven purely by circulation due to lift. Other force components on the wing have no influence on the wake. Therefore the model is not appropriate for simulating wakes behind bluff bodies, for example, where body forces are dominated by drag. Also, the inviscid flow model is appropriate for high Reynolds number flows such as those past large birds, but will be less accurate for modelling smaller flapping-wing animals, such as insects, where viscous effects become dominant.

- As vortex stretching is neglected, the model will not capture the increase in vorticity that arises on strained vortex filament segments (*Ananthan & Leishman, 2004*).

## RESULTS

For a pair of fixed, rectangular wings the wake rolls up downstream of both the inboard and outboard sections, as each wing acts as a separate lifting surface (Fig. 3A). For flying animals lift augmentation from the body would reduce the strength of trailing vorticity in the inboard sections. This phenomenon is readily modelled using the existing method by setting the circulation of the most inboard filaments to zero, which implicitly defines a uniform lift distribution across the body where the lift per unit span is equal to that of the wings (Fig. 3B). The geometry of the wake is visualised more clearly by applying a cubic spline interpolation of the marker positions (Fig. 3C).

A numerical convergence study for flapping-wing wakes was conducted by varying the three wake parameters: the number of timesteps per cycle, $n_t$, number of vortex filaments,

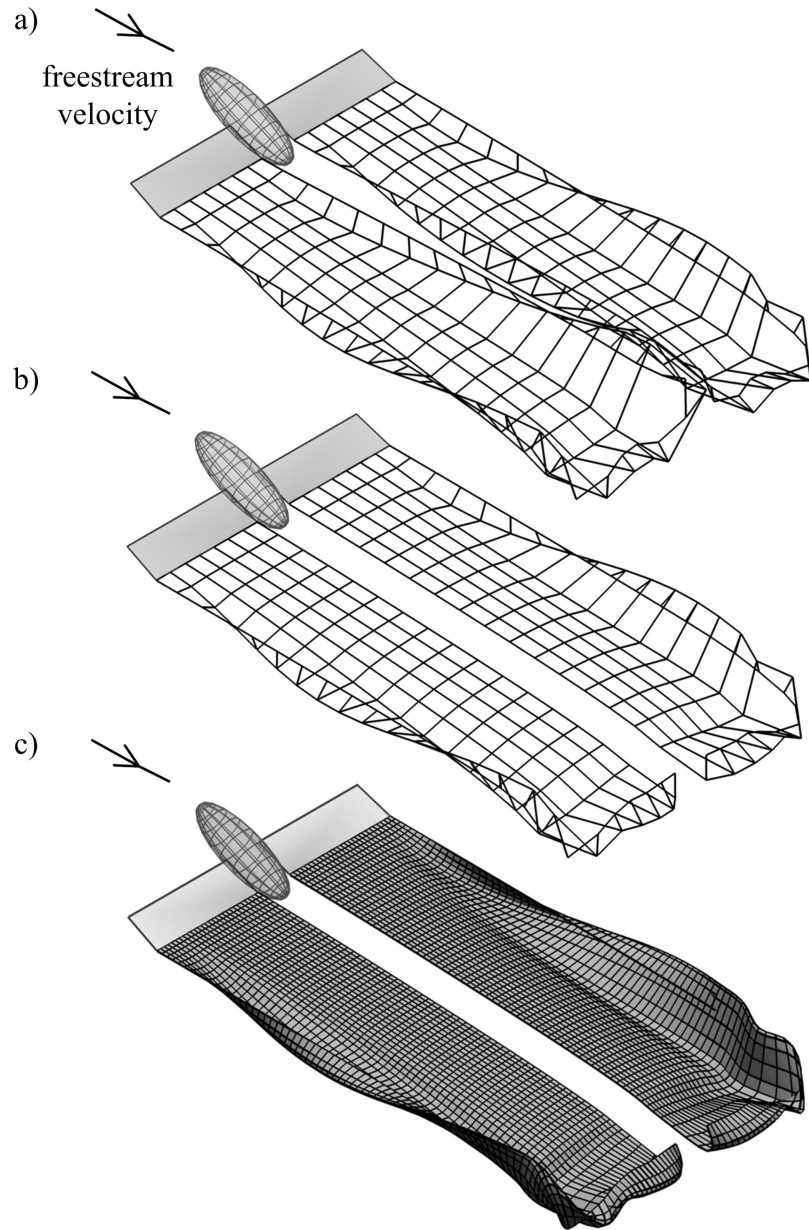

a)

freestream velocity

b)

c)

**Figure 3** **Simulated wakes for fixed rectangular wings in a 10 m s$^{-1}$ flow; wing length and surface area are the same as those of the Rock Pigeon.** (A) The influence of the body is neglected; (B) the body is assumed to have the same lift per unit span as the wings. (C) Simulation from (B) with a cubic spline interpolation of the Lagrangian marker positions.

$n_f$, and number of markers released per cycle for each filament, $n_m$. The normalised marker position, $p_{\mathrm{norm}}$, was used to measure the overall wake geometry:

$$p_{\mathrm{norm}} = \frac{\sum \|\mathbf{x}^{(n)}\|}{n_m n_f \|\mathbf{V}_\infty\| T};$$

(4)

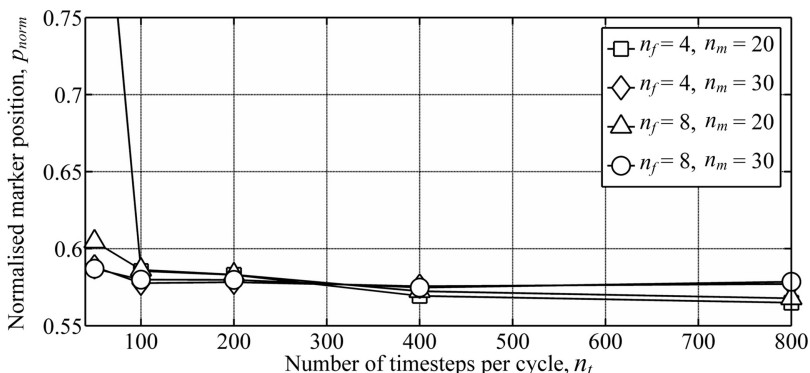

**Figure 4 Numerical convergence study showing the variation in wake geometry with varying numbers of the wake parameters.** Wake parameters include the number of time steps per cycle, $n_t$, number of vortex filaments, $n_f$, and number of markers released per cycle for each filament, $n_m$. Simulations were conducted using wing dynamics predicted from models of the Rock Pigeon in minimum power cruising flight (*Parslew, 2012*). The wake was simulated for three wingbeats, and the marker positions were taken from markers released during the third wingbeat to avoid interference from the starting vortex.

where $\|\mathbf{V}_\infty\| T$ is the distance travelled by the body over the period, $T$. Wing dynamics were taken from a model of the Rock Pigeon in minimum power cruising flight (cruising at $12 \text{ m s}^{-1}$, with a chord-based Reynolds number of $10^4$) (*Parslew, 2012*). Circulation of the inboard filaments was again set to zero to assume a uniform lift distribution on the body. The wake was simulated for three wingbeats, and $p_{\text{norm}}$ was derived for markers released during the third wingbeat. Extending the simulation to four wingbeats led to less than 1% change in $p_{\text{norm}}$. As with previous free-wake models (*Parslew, 2012*) the wake geometry converged when increasing the three wake parameters (Fig. 4). Doubling any of the three parameters from the maximum values shown in Fig. 4 led to less than 1% change in $p_{\text{norm}}$.

As expected from the lift distribution on a flapping wing, the wake rolls up downstream of the wing tips where the circulation strength is highest (Fig. 5). The decrease in lift from the wing tip to wing root gives negative circulation on the inboard trailing filaments. Thus, the wake rollup downstream of the inboard wing is of the opposite sense to the rollup downstream of the wing tip.

The wake geometry continues to evolve due to the self-induced velocity as it propagates downstream. This highlights physical instability in the wake [18], and is not the result of errors in the numerical method. Interaction between starting and tip vortices can be observed in the wake shed during the first wingbeat (Fig. 5C). This demonstrates the need to simulate the wake for more than one wingbeat to reduce the influence of the starting vortex on the overall wake geometry and thus obtain a steady-state solution.

Induced velocity can also be calculated on a mesh of nodes at fixed locations (Fig. 5D). This velocity field illustrates the counter-rotating wake regions downstream of the wing tips and roots, as seen in previous experimental studies of flying animal wakes (*Henningsson, Muijres & Hedenström, 2011*; *Hedenström et al., 2007*; *Muijres et al., 2008*;

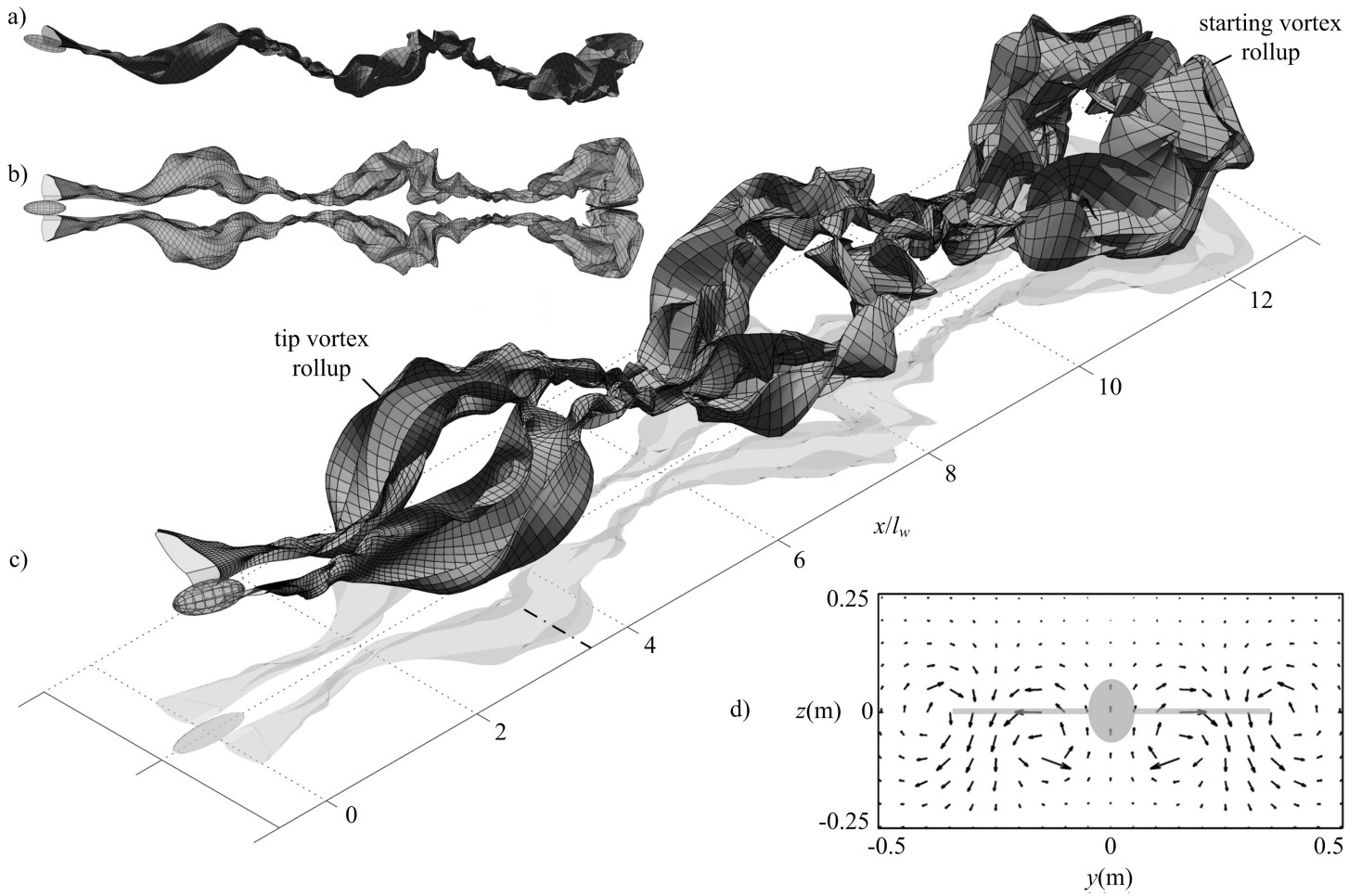

**Figure 5** **Wake geometry for simulated minimum power cruising flight of the Rock Pigeon over three wingbeats.** Wing dynamics were obtained using the model presented in *Parslew (2012)*. Simulations were conducted using 800 time steps per cycle, 8 filaments per wing, and 30 markers released per cycle for each filament. Wake surfaces are shown from side (A) top (B) and isometric (C) perspectives; dot-dash line at $x/l_w = 3.6$ indicates the streamwise location of the velocity plane used in (D), which shows the induced velocity predicted on a uniform grid of nodes when viewed in the negative $x$ direction, with a greyed projection of the bird body and wings.

*Hubel et al., 2010*; *Hedenström et al., 2009*; *Johansson & Hedenström, 2009*; *Bomphrey, Taylor & Thomas, 2009*). However, as vorticity is distributed on a discrete number of markers, reconstructed fields contain fluctuations in regions where the grid nodes are near to the markers. This led to the predicted induced velocity magnitudes being sensitive to the choice of plane streamwise location. For this reason the authors have chosen not to attempt to compare the computed velocity fields with those measured in previous flow visualisation experiments as this would be potentially misleading.

## DISCUSSION

The free-wake method is robust in that it can simulate wakes for different wing geometries and dynamics without requiring any changes to the underlying method. However, for extreme wing retraction a vortex core model is necessary to avoid numerical instabilities

(*Parslew, 2012*); when simulating the wake for cruising flight of the pigeon numerical instabilities were observed when the wing length on the upstroke was reduced to less than a quarter of its outstretched length.

Simulated fixed-wing wakes illustrated the need to define an appropriate lift distribution on the body to accurately predict the wake geometry downstream of the inboard wing. Conversely, flapping-wing wake geometries were insensitive to the model of body lift; wakes simulated with uniform or zero lift per unit span on the body contained no discernible differences in geometry. This is because the wake geometry is dominated by the rollup from the wing tip vortices, inboard wing vortices, and starting vortex, which occur for both models of body lift.

The starting vortex rollup caused significant differences in the wake geometry during the first wingbeat when compared to previous simulations that used a multiple horseshoe vortex model (*Parslew, 2012*). Despite this, wake geometries reached periodic solutions after three wingbeats. The computation time required to simulate the vortex ring model is significantly greater than for horseshoe vortex models, as the Biot-Savart law is evaluated for both trailing and shed vortex segments.

The free-wake method is useful for providing low order predictions of wake geometries for different wing geometries and dynamics. However, reconstructing the velocity field is more challenging and should be approached with some caution. Modelling the viscous vortex core has potential for preventing fluctuations in the induced velocity field, but this introduces additional uncertainty into the model in defining an appropriate vortex core radius.

### Funding
This work was funded by the Engineering and Physical Sciences Research Council (EPSRC) through the Postdoctoral Fellowship Prize. The funders had no role in study design, data collection and analysis, decision to publish, or preparation of the manuscript.

### Grant Disclosures
The following grant information was disclosed by the authors:
Engineering and Physical Sciences Research Council (EPSRC).

### Competing Interests
The authors report no competing interests.

### Author Contributions
- Ben Parslew conceived and designed the model, performed the simulations, analyzed the data and wrote the paper.
- William J. Crowther conceived and designed the model, analyzed the data and wrote the paper.

**Peer**J ________________________________

## Data Deposition

The following information was supplied regarding the deposition of related data:
Figshare http://figshare.com/articles/Theoretical_Modelling_of_Wakes_from_Retractable_
Flapping_Wings_in_Forward_Flight/729051.

## Supplemental Information

Supplemental information for this article can be found online at http://dx.doi.org/
10.7717/peerj.105.

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
