# Peer review of "Theoretical modelling of wakes from retractable flapping wings in forward flight"

_PeerJ, doi:10.7717/peerj.105_

## Round 0.1 · original submission · Minor Revisions

I am very pleased to report that both reviewers have done open reviews (please use the option to publish the reviews with your paper!) and they are positive, so we can provisionally accept the paper as long as minor revisions are finished. Send back a revised MS and I will check it to ensure these are satisfactory-- I think all the suggestions are very reasonable and achievable.

It is extremely important to make the data for this study available (I recommend Figshare or Dryad) to maximize the openness of the study, in the spirit of PeerJ. However, minimally I ask that a clear statement of how to obtain the data is made. "Data can be obtained from the author on request" is common but this is not truly open science. Regardless, I leave this decision to the author, as practice within the field currently varies.

·

Basic reporting

The article meets reporting standards.

Experimental design

The MS is interesting and clearly written. I am not a fluid dynamicist, but the methodology is clear and seems technically sound. Briefly, instantaneous local circulation follows from the Kutta-Joukowski theorem, using local lift, from blade element theory. The circulation is used to calculate local induced velocity using the Biot-Savart law. The local flow velocity, following from the free stream and induced velocity, is then integrated to obtain local flow geometries.

Validity of the findings

The manuscript introduces a free-wake method to model time-varyingly the wake geometry of simulated pigeon cruising flight. The free-wake method, which does not require resolving the entire flow field, is applied to simulated retractable, jointed flapping wings, and is evaluated for sets of wake parameters. The conclusions are appropriate and supported by the results.

Additional comments

What follows are some minor comments:

1) For a general journal it may help to define 'numerical convergence', to avoid a potential misunderstanding that the model and actual pigeon wakes are shown to converge.

2) Abstract, line 4: “The free-wake model is robust in simulating a range of wing geometries and dynamics …”. Unclear. Simulating wake geometries for a range of wing geometries and dynamics?

3) Page 3, line 50: Please define "transverse" (i.e. specify whether wrt wing chord or span)

4) Page 4, line 70: Please provide a reference for blade-element theory.

5) In your limitations, please also address effects of the absence of unsteady aerodynamic forces in your model ( e.g. acceleration reaction forces)

6) Since computational speed is one of the advantages of the model, can you provide a relative quantification of the computational load?

7) Are empirical data available to which your results can be compared (e.g. wake PIV measurements)? If not, please mention. If so, how do the results compare?

8) Page 6, line 109 what was the modeled flight speed (pigeon min power cruising flight) and Re?

·

Basic reporting

This is a well-written manuscript with a clear motivation and set in good context within the field.
The five figures augment the text and are both informative and attractive.

Some references to prior literature could be added:

L12 References 6-8 are rather limited and unbalanced. Others could be added - e.g. Ellington, C. P. The aerodynamics of hovering insect flight. V. A vortex theory. Phil. Trans. R. Soc. Lond. B 305, 115-144 (1984).

L14 Ref 9 is not the first attempt at this and seems to be chosen arbitrarily.

L70 Do these references contain modifications/adaptations of standard blade element theory? If not, they are perhaps not the most appropriate to references to use here.

L127. Ref 17 is not the only example of root vortices in the wakes of animals. It has also been seen behind e.g. swifts and bats (which are even more relevant to the model you present here).

Experimental design

The aim of the manuscript is relevant and timely and future refinement of the model (as described at the end of the Discussion) will increase its utility further. The investigation has been conducted with rigor and to a high technical standard. The method is reproducible on using the information given in the text.

Validity of the findings

The data appear to be sound and robust to variation in the model parameters (Fig. 4) within the limitations of the model (highlighted by the authors in section 2.3).

No reference is made to the availability of the data.

Additional comments

Comments that might improve the clarity and accuracy of the manuscript:

L25. add 'of' after 'number'.

L56. Should this be "across the span" instead of "throughout the wingbeat"?

L68. "only feasible for a small number of markers". Could you provide an order of magnitude that represents a feasible" number? Would this be using high performance computing or a single PC?

L84. "velocity field immediately downstream ... will not be accurately resolved". It might be useful to include a sentence of discussion on this point - perhaps including an estimate of the potential error in this case, or the conditions under which the error is likely to me most significant.

L108. There is a transition here from discussion of a Gliding model to Flapping flight model that could be stated more clearly.

Figure 5. What model of circulation over the body was used for this simulation?

L119. This evolution of the wake geometry under self-induced velocity is discussed in Bomphrey, R. J., Henningsson, P., Michaelis, D. & Hollis, D. Tomographic particle image velocimetry of desert locust wakes: instantaneous volumes combine to reveal hidden vortex elements and rapid wake deformation. Journal of The Royal Society Interface, doi:10.1098/rsif.2012.0418 (2012).

L133. What would constitute 'extreme' wing retraction?

---

## Round 0.2 · accepted · Accept

This is a wonderful contribution to PeerJ and I am very glad you've submitted it here. The peer review process has also worked very well to produce an improved final product. I am happy to accept this as-is. Well done!

Please tick the option to publish the reviews/revisions with the paper! This paper is a great example of where open review has worked nicely. The MS will be augmented by the publication of the review process so anyone can see. But of course this is your decision in the end. I wish to thank the 2 reviewers for being non-anonymous, though.